# Cyclone vulnerability assessment of the central coast of Bangladesh: A comprehensive study utilizing FAHP and geospatial techniques

**Md Abdullah Salman** [ID]\*, **Mahmudul Hasan Rakib, Shacin Chandra Saha**[ID],
**Md Emdadul Haque, Md Sabbir Hossen**[ID]

Department of Geology and Mining, University of Barishal, Barishal, Bangladesh

\* masalman@bu.ac.bd

## Abstract

The coastal areas of Bangladesh are recognized as a major South Asian center for cyclone landfall. This research develops a comprehensive tropical cyclone mapping strategy utilizing the Fuzzy Analytical Hierarchy Process (FAHP) and geospatial techniques to analyze the vulnerability distribution in the central coastal regions of Bangladesh. Eighteen spatial features, categorized into physical, social, and mitigation capacity criteria, were assessed to evaluate vulnerability. The output indicates that the southern peripheral districts- Bhola, Borgona, and Patuakhali are more vulnerable to tropical cyclones due to factors such as historical cyclone tracks, proximity to the coastline, low elevation, gentle slopes, high population density (including vulnerable groups such as females, the disabled, and agricultural workers), poor socioeconomic status, and land covers (crops and vegetations) prone to damage. Mitigation measures in these areas, including cyclone warnings, embankments, and access to shelters and road networks, are found to be inadequate. Validation through ROC and AUC confirms the accuracy of vulnerability maps. These findings offer critical insights for policymakers, local NGOs, and local administrators to enhance cyclone preparedness and develop targeted mitigation strategies to reduce vulnerability in coastal Bangladesh.

## Introduction

Tropical cyclones exhibit distinct features like reduced air pressure, intense winds, storm surges, and substantial precipitation and are widely recognized as the most formidable weather events with significant destructive potential on a global scale [1]. These natural disasters frequently result in high death tolls, extensive property and agricultural losses, and interruptions to infrastructures and communication networks [2]. In all tropical cyclone basins, the quantity and percentage of intense tropical cyclones have grown [3]. Over time, numerous 'climatologies' have been created to record

**Data availability statement:** All relevant data are in the paper.

**Funding:** The author(s) received no specific funding for this work.

**Competing interests:** The authors have declared that no competing interests exist.

the occasions, locations, and environmental circumstances in which cyclones occur [4]. Tropical cyclone intensities may change as a result of changing environmental variables, and internal variations that could signal storm-scale instabilities or the stochastic effects of high-frequency transients like moist convection may reflect these changes [5]. Eyewall cycles in cyclones may be started by environmental factors, but once they start, they continue on their own [6]. Tropical cyclones have caused significantly more economic damage globally in recent decades [7]. Many coastal areas around the world frequently experience the effects of tropical cyclones. On average, seven tropical cyclones hit the mainland of China and Hainan Island every year, causing direct economic losses of approximately 28.7 billion Yuan and 472 fatalities [8]. This vulnerability could become more acute as a result of sea level rise [9].

Bangladesh is currently regarded as one of the nation's most prone to natural disasters. Floods, storm surges and cyclones, flash floods, droughts, riverbank erosion, and landslides are the key calamities that are at threat here [10]. Typically, every year, either in the early summer (April–May) and/or the late rainy season (October–November), cyclones strike Bangladesh's coastal areas. Significant portions of the coastline are damaged by at least one major cyclone at least once every three years [11]. Cyclones feature severe wind velocities and heavy precipitation [12]. Devastating consequences can result from storm surges, which are characterized by abnormally high sea levels, such as loss of life and property. Flimsy housing is frequently dislodged or severely damaged by winds that exceed speeds of hundred kilometers per hour [13]. Intense precipitation is characterized by high volumes of rain within a brief time frame. Prolonged rainy periods could lead to severe floods and significant damage to crops. Like milder cyclones, flooding of this nature can occur. Additionally, if reservoirs collapse and landslides occur, it can result in a greater amount of deaths and destruction of property [14].

A destructive cyclone and tidal surge from the Bay of Bengal slammed Bangladesh's southern coastal areas during the night of April 29 and early hours of April 30 in 1991, affecting 5 million people and all the districts of the country [15]. In addition, around 300,000 lives were lost due to the 1970 Bhola cyclone [14]. The high level of human impact in this area results from both significant human presence and the dominant influence of human activity. The prevailing socio-cultural beliefs and practices, as well as the unfavorable economic circumstances, affect the majority. The absence of a cohesive institutional strategy for disaster planning and management has left vulnerable coastal communities unprotected [16]. The housing sector lost BDT 57.9 cores due to the destructive effect of super cyclone Sidr [17]. The incomprehensible quantities of casualties caused by cyclonic activity in Bangladesh have resulted in a significant number of fatalities, with 300,000 fatalities accounted for in 1970 and 5704 fatalities in 1988, respectively. The total number of fatalities in 1991 was recorded at 138,866, while the number of deaths significantly reduced to 4234 in 2007 [18].

Prior research on disasters in Bangladesh has primarily focused on riverine hazards [19]. In addition to information on riverine hazards, there is literature on cyclones and storm surges in Bangladesh. Some have concentrated on numerical modeling and storm surge predictions [20]. There are a few research studies on the causes,

consequences, and approaches to mitigation for cyclones and storm surges [21–23]. Paul [24] explores the change in cyclone disaster vulnerability and response in coastal Bangladesh, identifying social transformation and risk perception as key elements of resilience. Additionally, studies by Paul and Dutt [25] on factors of cyclone disaster deaths in coastal Bangladesh and Alam and Collins [26] on non-evacuation and shelter-seeking behavior provide critical perspectives on human responses, which are essential for understanding social vulnerability and formulating effective mitigation strategies. These references further enrich the contextual grounding of this study and highlight the importance of integrating behavioral, cultural, and systemic factors into spatial vulnerability assessments.

The selection of acceptable and sufficient criteria, scale, and risk components are what determine the accuracy and level of detail of the information regarding the potential for tropical storm damage [27]. A more reliable risk assessment can be built on a foundation that consists of the adequate selection of criteria for each of the risk components (such as vulnerability, hazard, exposure, and mitigation), as well as the processing of those criteria [28]. In order to effectively identify the kind of risk and select the most effective risk mitigation measures as part of the method of developing suitable management plans, it is necessary to have detailed information about risks at the local scale [29]. Vulnerability, in essence, pertains to the extent to which an individual or property is susceptible to harm or attack [30]. Sims and Baumann's [31] discussed alert systems for hazards and readiness initiatives. Communicating information through this method has proven to be successful in developed countries for educating individuals to prepare for potential harm and adjust ways of utilizing resources [32]. There have only been a few studies that take into account geospatial technologies and natural disasters [18].

Implementing effective measures to mitigate risks can significantly minimize the negative consequences and ramifications of tropical cyclones [33]. Efforts utilizing Geographic Information System (GIS) technology have been explored to evaluate the danger of cyclone susceptibilities and the ability to reduce potential harm in Bangladesh [34]. The spatial data management framework of GIS is highly robust, enabling it to demonstrate exceptional performance, especially the ability to adeptly examine and combine diverse sets of information with versatile methods to evaluate and understand [35]. The current or dominant system of Effective management of disasters in Bangladesh holds significant relevance to those engaged in similar efforts [36,37]. Sophisticated mapping methodologies that take multiple factors into account are regarded as superior because they furnish in-depth data on the spatial susceptibility to tropical cyclone repercussions [38]. According to the literature, the Fuzzy Analytical Hierarchy Process (FAHP) is the most prevalent, widely recognized in risk assessment associated with cyclones [39] and well-suited approach for spatial multi-criteria evaluation [40]. Compared to AHP, FAHP is more effective at dealing with the ambiguity and uncertainty of human judgments in pairwise comparisons, making it particularly valuable in disaster risk assessment where expert opinions may not always be precise.

Although numerous studies have been conducted on cyclone vulnerability assessment in Bangladesh, this study presents a novel contribution by focusing specifically on the middle coastal zone, which has received less attention in previous research compared to the eastern and western regions. Furthermore, it integrates both physical and social vulnerability along with mitigation capacity using FAHP, which enhances decision-making accuracy under uncertainty better than traditional AHP methods.

This study aims to establish a thorough approach to mapping the vulnerability of tropical cyclones, one that takes into account both physical and social vulnerability, as well as the capacity for mitigation. The suggested methodology in the central coastal zone of Bangladesh involves examining the geographical distribution of vulnerability to tropical cyclones. The study is conducted with a specific focus on three primary objectives:

1. To assess an index of tropical cyclones' physical and social vulnerability and capability for mitigation utilizing FAHP and geospatial methods.

2. To evaluate a vulnerability map that analyzes the geographical pattern of susceptibility to tropical cyclones. This assessment incorporates many indices that measure both physical and social vulnerabilities, along with the capacity for mitigation.

3. To verify the accuracy of the results obtained about spatial vulnerability.

## Methodology

### Study area

The Ganges Brahmaputra Meghna (GBM) river system and the Bay of Bengal influence the geomorphology and hydrology of Bangladesh's coastal region. About 32% of Bangladesh's total landmass, or 47,201 km², is made up of the coastline region, which includes 19 districts [41]. According to geographical features, Bangladesh's coastal zone is classified into three sections: (a) the western zone, (b) the central zone, and (c) the eastern zone (Ahmad, 2019) (shown in Fig 1). However, to preserve coherence, the study only chose five districts that are prone to cyclones and their 46 Upazilas in the central coastal zone, which has an area totaling 8975.51 sq. km. The region has a population of 9,100,102 with a density of 690 per km² [41]. Geographically, Bangladesh's central coastline region is located between 21°50'0" and 23°50'0" N latitude and 89°50'0" and 91°0'0" E longitude (see Fig 1). Severe natural catastrophe vulnerability, which results in fatalities, destruction of economic and infrastructure assets, and detrimental effects on livelihoods, especially for the poor,

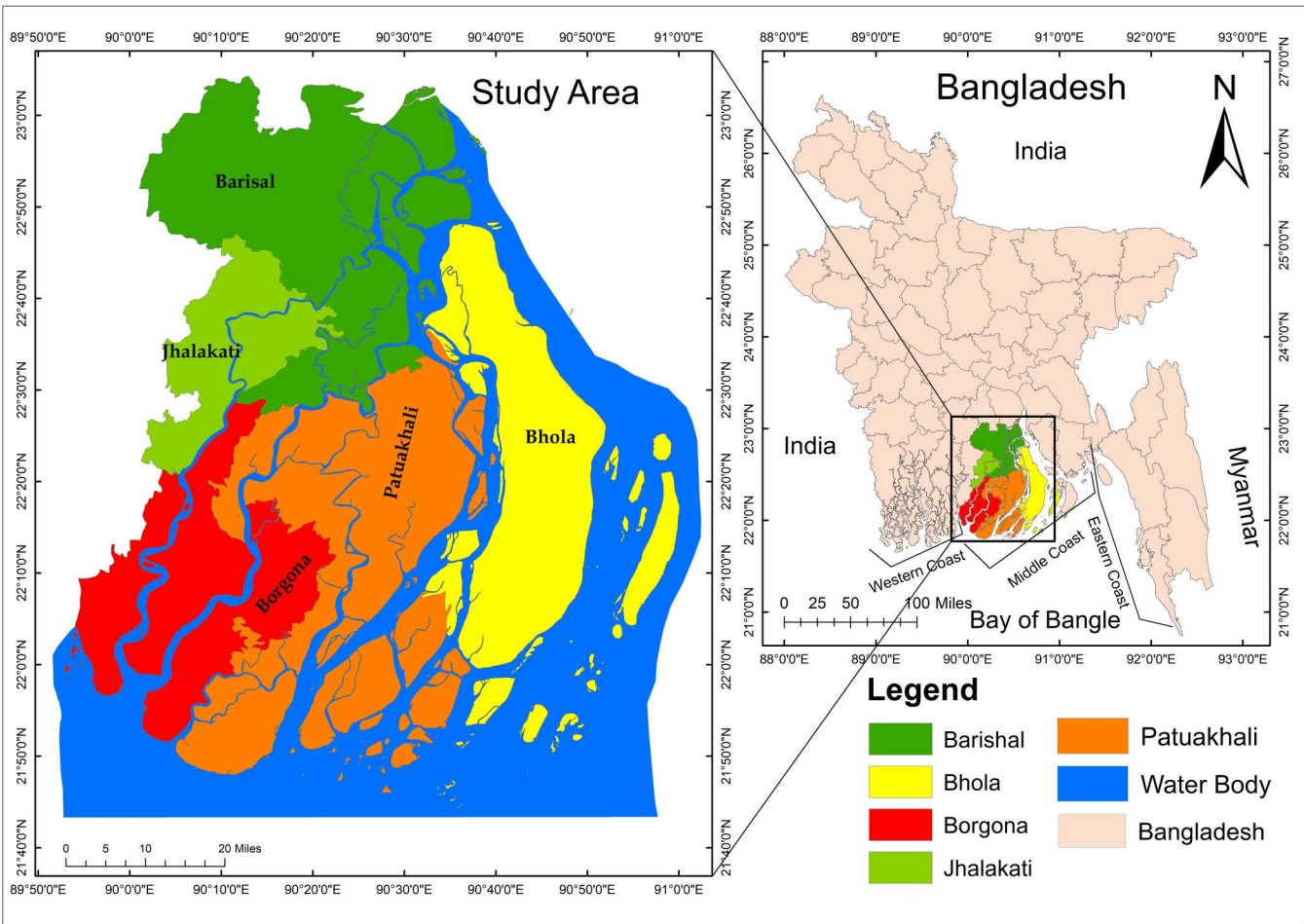

**Fig 1. Spatial distribution of the study area in the context of Bangladesh's coastal zones.** The left panel highlights the specific administrative districts under investigation—Barishal, Bhola, Patuakhali, Barguna, and Jhalokathi—which are among the most cyclone-prone regions in the country. The right panel delineates the broader coastal zonation of Bangladesh, categorizing the region into Western, Central (Middle), and Eastern coastal zones based on geographic and administrative boundaries. These zones provide a framework for assessing regional vulnerability and disaster impact variation along the coastal belt. [Satellite image courtesy of the U.S. Geological Survey (USGS)].

vulnerable, and impoverished living in environmentally exposed places- Particularly Bangladesh's southern coastline region. The geographical area under consideration has a humid climatic condition characterized by an average annual precipitation of 1955 millimeters [41]. Super Cyclones Sidr (2007) and Aila (2009) have severely damaged the coast, killing 3,500 people, injuring 191, and causing significant environmental and property damage [33]. To add, many still perceive them as a major cause of socioeconomic vulnerability and water scarcity in Southwestern Bangladesh [42]. The potential impacts of cyclones may extend beyond the immediate research location in an east-west direction. Notably, this study encompasses five coastal regions (Barishal, Jhalakhati, Borgona, Patukhali and Bhola) (shown in Fig 1) that have seen significant devastation from previous tropical cyclones.

## Methods

In this work, an FAHP-based multifaceted appraisal approach was employed to integrate social, physical, anthropogenic, and relief capacity factors to evaluate susceptibility to tropical cyclones. Several vulnerability conditions are outlined in the scientific literature for assessing susceptibility to natural hazards. Disaster hazard and vulnerability evaluation depend on different variables, such as fitting hypothetical concepts and the quality and adequacy of data accumulated [43]. Over time, a wide variety of equations have been used by scientists to evaluate the vulnerability of natural disasters. In this study, to calculate cyclone vulnerability, this study used Eq. (1) based on a literature review [27]. Fig 2 represents the systematic procedure of vulnerability assessment of the current study areas.

$$Vulnerability = \frac{Physical\ vulnerability \ \times \ Social\ vulnerability}{Mitigation\ capacity}$$

(1)

## Dataset

The assessment of cyclone hazard may be influenced by various factors [38]. Various forms of information were gathered from diverse origins and utilized for distinct intents. Different types of public domain data sources, for instance, domestic and foreign, were utilized in the study (see Table 1). The scope of this endeavor includes both governmental and private establishments in addition to on-site research activities. Over the course of one year, from 2022 to 2023, several field investigations were conducted to gather and assess data pertaining to the capacity for mitigation as well as validation information. No permits were required for this study as research was conducted in publicly accessible areas where no specific permissions were necessary. The fieldwork adhered to all applicable local regulations and guidelines. The GIS tool and spatial statistics were utilized to create layers for the assessment factor in a GIS setting. Data were gathered from various sources, both public online and offline domain (see Table 1).

## Vulnerability assessment

The choice of criteria and sub-factors in this study was informed by a comprehensive examination of the literature review and their relevance in assessing a region's vulnerability to tropical cyclones. The study created a geospatial system comprising physical, social, and mitigation capacity aspects, resulting in 18 spatial criteria layers. These layers were developed to assess cyclone vulnerability. The spatial resolution of every raster layer measures 30 sq. meters, which was used for geospatial analysis and imagery processing. The natural break categorization approach categorized the created maps because it was determined that it was more effective in this study at visualizing the spatial pattern of vulnerability [44] This section provides an overview of the mapping methodology employed and discusses the ramifications of the chosen criteria.

   **Assessment criteria for physical vulnerability.** Risk assessment and analysis rely heavily on the appropriate selection of factors, categorization into various categories, and mapping of various risk factors [38]. In the context of this research, specific physical phenomena and traits that influence the vulnerability of an area to tropical cyclones were chosen. These include land cover, slope, elevation, closeness to the cyclone track and shoreline.

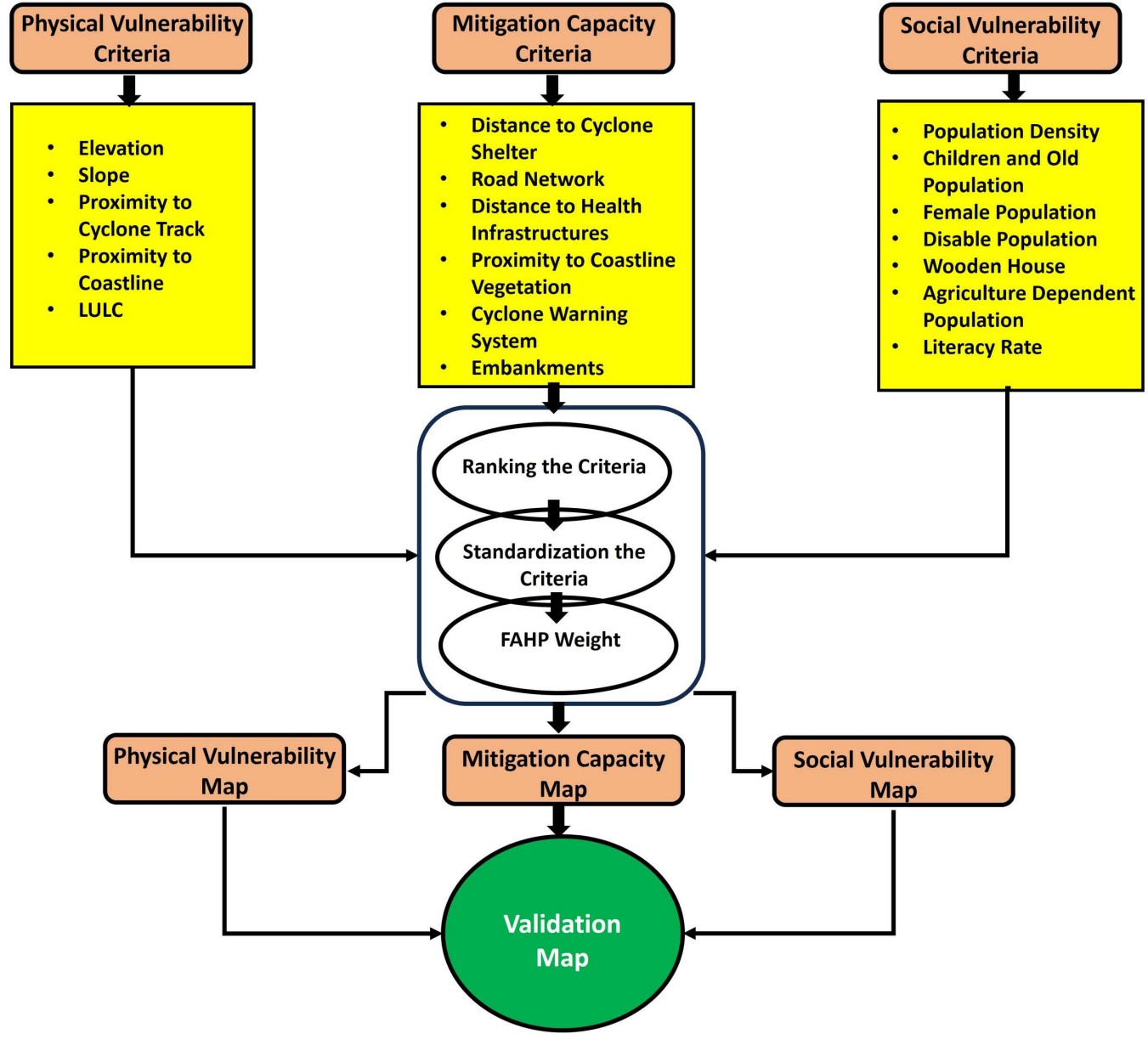

**Fig 2. Flow diagram of the FAHP-based multi-criteria decision-making framework for vulnerability assessment.**

In order to build land cover spatial layers, we made use of five Sentinel-2 and one Landsat-8 picture as specific types of land coverings are far more prone to damage from tropical storms than others [45] (see Fig 3). After applying the necessary pre-processing corrections to the satellite images, for instance, radiometric, geometric, and atmospheric corrections, a hybrid classification strategy was employed to determine the different land cover groups (see Fig 3a).

Unsupervised classification was employed at first to determine the most likely categories, and after that, supervised classification with a maximum likelihood method was utilized [46]. Extremely high levels of resolution in order to determine how accurately the land cover map represents terrain, 457 random points were generated with Google Earth Imagery,

**Table 1. Summary of datasets and their specific characteristics utilized in this study.** The table outlines the spatial, temporal, and thematic attributes of each dataset employed for analyzing the vulnerability and impacts of cyclones in the central coastal districts of Bangladesh. These datasets encompass satellite imagery, administrative boundaries, demographic data, and historical cyclone records, providing a comprehensive foundation for geospatial analysis and risk assessment.

| Types of Data | Data Sources | Time Period | Intended Use |
|---|---|---|---|
| SENTINEL-2 Imagery (10 m resolution) | USGS EROS (Earth Resources Observatory and Science Center) | 2022 | Land cover |
| Landsat 8 Imagery (30 m resolution) | USGS EROS (Earth Resources Observatory and Science Center) | 10 January, 2023 | Analysis of the vegetation along the coast |
| Digital Elevation Model (DEM) data (20 m resolution) | Survey of Bangladesh (SOB) | 2014 | Elevation and slope analysis |
| Population data | Bangladesh Bureau of Statistics (BBS) | Population census 2021 | Density of population, percentage of women in the population, prevalence of wooden structures, percentage of people with disabilities, literacy rate, and reliance on agriculture for sustenance. |
| Cyclone track data | International Best Track Archive for Climate Stewardship (IBTrACS) | 1960–2020 | Frequency of Cyclone, closeness to cyclone track |
| Road and Embankment data | LGED | 2018 | Embankments and Road network. |
| Health facilities, cyclone warning systems, and cyclone shelter | LGED and Fieldwork | 2018 | Location of cyclone shelters from the disaster-prone areas, medical facilities, and early warning systems |

2022 and 2023. Following that, an accurate evaluation approach that was based on the literature was carried out. After further investigation, the total accuracy was 90.27%, while the Kappa coefficient was found to be 88.94%.

Locations that are high in elevation and have steep slopes are likely to have a lower risk of being affected by natural disasters. Besides, locations that are low in elevation and have a moderate slope, on the other hand, are labeled as having a greater vulnerability [47].

The slope and elevation maps were produced with the use of a digital elevation model (DEM) that has a spatial resolution of 20 meters (see Figs 3b,c). The DEM was created with the use of the topographic sheet that was obtained from the Survey of Bangladesh. The sheet had a scale of 1: 25,000. The DEM has an accuracy of b/ 50 cm in the Z dimension.

In addition, lives and properties that are positioned close to historical storm tracks and the shoreline are at an increased risk of being affected by tropical cyclones [14]. As a result, characteristics such as closeness to cyclone tracks and shorelines were taken into consideration while evaluating a location's potential physical vulnerability to tropical cyclones. From the cyclone track datasets, 41 spatial cyclone tracks, encompassing cyclones of varying intensities from category one to category five, were identified across the research region between the years 1960 and 2020, and they were included in the buffer analysis (see Fig 3d). On the other hand, distances from the shoreline of the research region were determined by making use of the ruler tool that is included in Google Earth Pro [48]. After that, the spatial layer representing the distance from the coast was created [49].

**Assessment criteria for social vulnerability.** Individuals, collectives, or entire communities that exhibit deficiencies in various aspects, such as substantial educational attainment, the establishment of security and tranquility, the enjoyment of fundamental human rights, the presence of a stable political framework, an equitable allocation of resources, robust social conventions, or a profound faith in a transcendent entity, are rendered more vulnerable to the adverse consequences arising from natural calamities. The allocation of aid funds often exhibits a tendency to favor religious, political, and social entities while simultaneously neglecting the needs of women, destitute individuals, and religious minorities. Consequently, this prevailing situation predominantly places women at a disadvantage [50]. Therefore, the characteristics and criteria

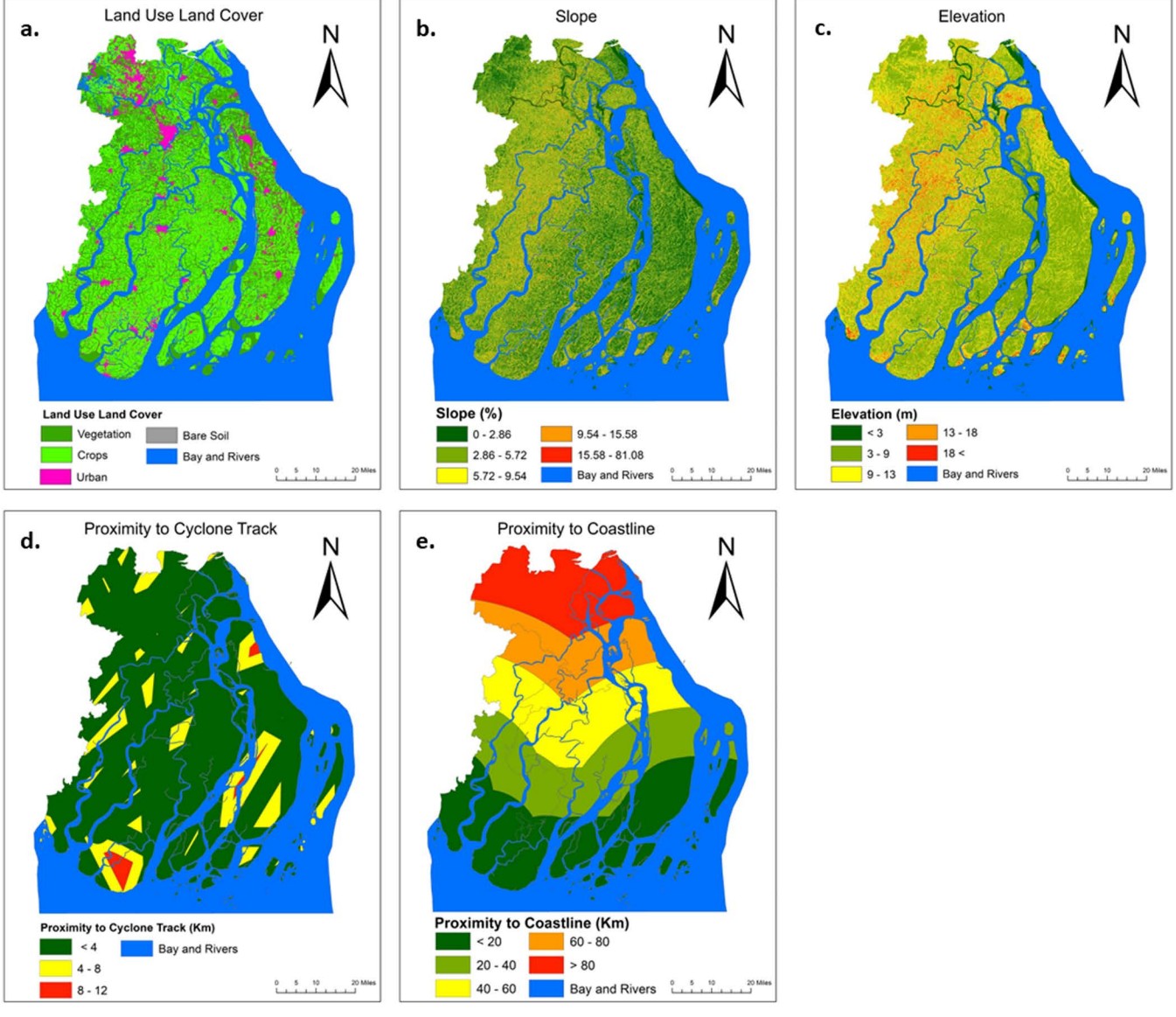

**Fig 3. Feature for physical vulnerability.** (a) land cover, (b) slope, (c) elevation, (d) proximity to cyclone track, and (e) proximity to coastline. [Land cover map created using SENTINEL-2 Imagery (10 m resolution) from the U.S. Geological Survey (USGS)].

that were chosen, such as the population density, the population of women, the population of people with disabilities, the population of people without jobs, the percentage of individuals whose livelihoods are dependent on agriculture, the literacy rate, and the percentage of houses made of wood (see Fig 4). To create all the necessary spatial layers for social characteristics, this study utilized the most recent available housing and population census data (2021) run in Bangladesh every ten years.

Population density is a factor that highly determines the degree of damage a particular area will experience. Being crisscrossed by rivers, the central part of Bangladesh provides fertile lands, easy ways of transportation, and suitable ports, attracting people to settle here from earlier times [51]. Therefore, the central part of Bangladesh and its coastal

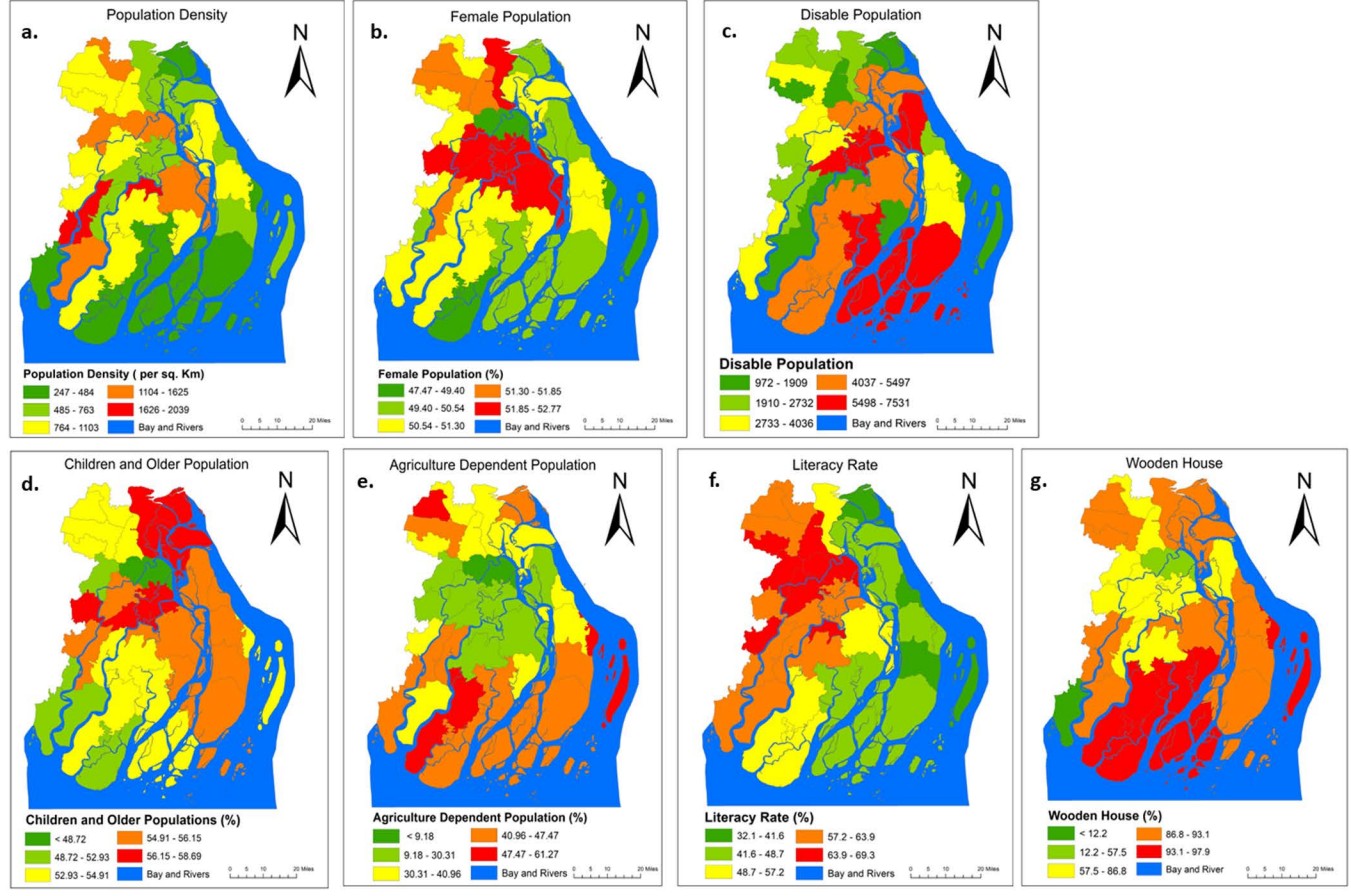

**Fig 4. Feature for Social vulnerability.** (a) population density, (b) female population, (c) disabled population, (d) dependent population, (e) agriculture-dependent population, (f) literacy rate, and (g) wooden house. [Satellite images courtesy of the U.S. Geological Survey (USGS)].

component has a population of moderate to high density. More people reside in areas more vulnerable to tropical cyclone damage (see Fig 4a). Children and the elderly are specifically vulnerable to the impacts of tropical cyclones because of their reduced capacity to evacuate safely and independently in the face of an emergency [52]. This study utilized data from the 2021 population and housing census conducted by the Bangladesh Bureau of Statistics (BBS) [53] in order to generate the population density, female, disabled, children, and elderly population geographical layers (see Figs 4a–4d).

A physically fit human is more strongly equipped with facilities to survive a disaster than an unfit or disabled human. A disabled individual suffers more during a natural disaster because he/she has restricted mobility, expertise, access to resources, and other obstacles, including security, before a catastrophic emergency [54].

According to cultural norms, women are responsible for the management of family finances and must obtain permission from their husbands before entering a safe house, which prohibits the women from attending cyclone risk reduction training programs and going to cyclone shelters [26]. In this investigation, the study created the spatial layers for the female population and the disabled population using the most recent census data (see Fig 4b). Nearly every year, tropical cyclones bring significant devastation to the agricultural areas of countries located in the southern part of the world. It is of the utmost importance to determine the extent of the damage to the rice-growing agricultural land regions in the countries

of South and Southeast Asia [55]. Because of this, people whose livelihoods depend on agriculture are at risk of suffering significant financial setbacks as a result of the effects of tropical cyclones (see Fig 4e).

Education is a crucial indication that has a considerable influence on people's susceptibility. It was demonstrated that those with lower levels of education were less capable of comprehending the catastrophe prognosis and determining the proper quantities of food and supplies to stockpile in case of evacuation [56]. The literacy rate geographical layer that was used in this investigation was prepared with the help of the housing and population census data from 2021 (see Fig 4f).

The most important factors in determining a household's level of vulnerability are the house's construction materials and its location. In coastal Bangladesh, many people live in one of four distinct styles of dwellings (see Fig 4g). The Pucca Houses are the most stable and long-lasting homes since they are constructed out of brick and concrete. However, these buildings are extremely uncommon and are only owned by wealthy families. The walls of the semi-pucca structures are often built of bamboo mats or lumber, whereas the roof consists of corrugated iron (CI) sheets. The floors of the semi-pucca buildings can be made of mud, brick, or cement. Mud is used to construct the floors of kutcha homes, which have walls made of bamboo, twigs, or straw, and roofs composed of wheat straw or paddy (rice) [57]. Since the building material Semi Pucca and Kutcha houses are dominated by woody material, we categorized them as woody houses, and they are more susceptible to damage by disaster than Pucca houses. It is a terrible condition when houses are situated along the embankments since they are at risk of suffering catastrophic losses and damage if there is a storm with high winds and waves or if there is any kind of breakdown in the dikes [58]. In order to produce the agriculturally dependent geographical layer, we first divided the percentages of agricultural dependence into five distinct categories. The information that we needed to analyze either of those variables was taken from the housing and population census data of 2021.

**Assessment criteria for mitigation capacity.** Even though technology is always getting better at predicting, finding, and describing the physical dimensions of tropical storms, as well as sending warning messages and keeping an eye on their progress, this loss keeps going up at an alarming pace [59]. The ability to mitigate the risks that tropical cyclones pose is a mirror of the main plan, which could include both structural and non-structural steps to make it less likely that tropical cyclones will cause damage [60].

Since the late 1960s, plans have been in place for East Pakistan to build cyclone shelters. Shelter construction became the mainstay of cyclone damage prevention strategies because it helped save many lives in the wake of the 1991 storm [61]. Cyclone shelters have been acting as a major structural platform to save the life in rural Bangladesh. Similarly, the presence of health infrastructures is also a factor that affects the vulnerability of a particular region. During cyclone events, health infrastructures facilitate the provision of vital emergency medical care to impact individuals [38]. In order to conduct this study, data pertaining to cyclone shelters was obtained from the Local Government Engineering Department (LGED). The accuracy and reliability of these findings were ensured by many trips to the field spanning from June 2022 to June 2023. Subsequently, the Euclidean distance approaches were employed to assess the distances and derive the proximity between many layers of cyclone shelters and health services (see Figs 5a, 5b).

The force of wind and storm waves, in addition to the speed of currents, can be mitigated by mangroves and other coastal trees. In addition to these benefits, these forests defend the coastline from erosion and keep wetland areas intact. For every kilometer of forest that is there, a dense forest can minimize the height of the surge by half a meter [62]. The spatial vegetation cover data obtained from the Bangladesh Forest Department were utilized in this study to determine the locations within the study site that were covered by coastal vegetation. In order to calculate the distance from the study area to the coastal vegetation, a tool in ArcGIS named 'Euclidean distance' was used. These tools were based on the coastal vegetation that was detected in the study region (see Fig 5c).

In most cases, flood mitigation from embankments, levees, and dikes is provided up to a certain level of severity [63]. Embankments protect the lands of agriculture, homesteads, and dwelling places from storm surges. It also prevents seawater intrusion. In the following study, the spatial layer of the embankment was generated by using the LGED data to determine whether or not there were embankments present (see Fig 5d).

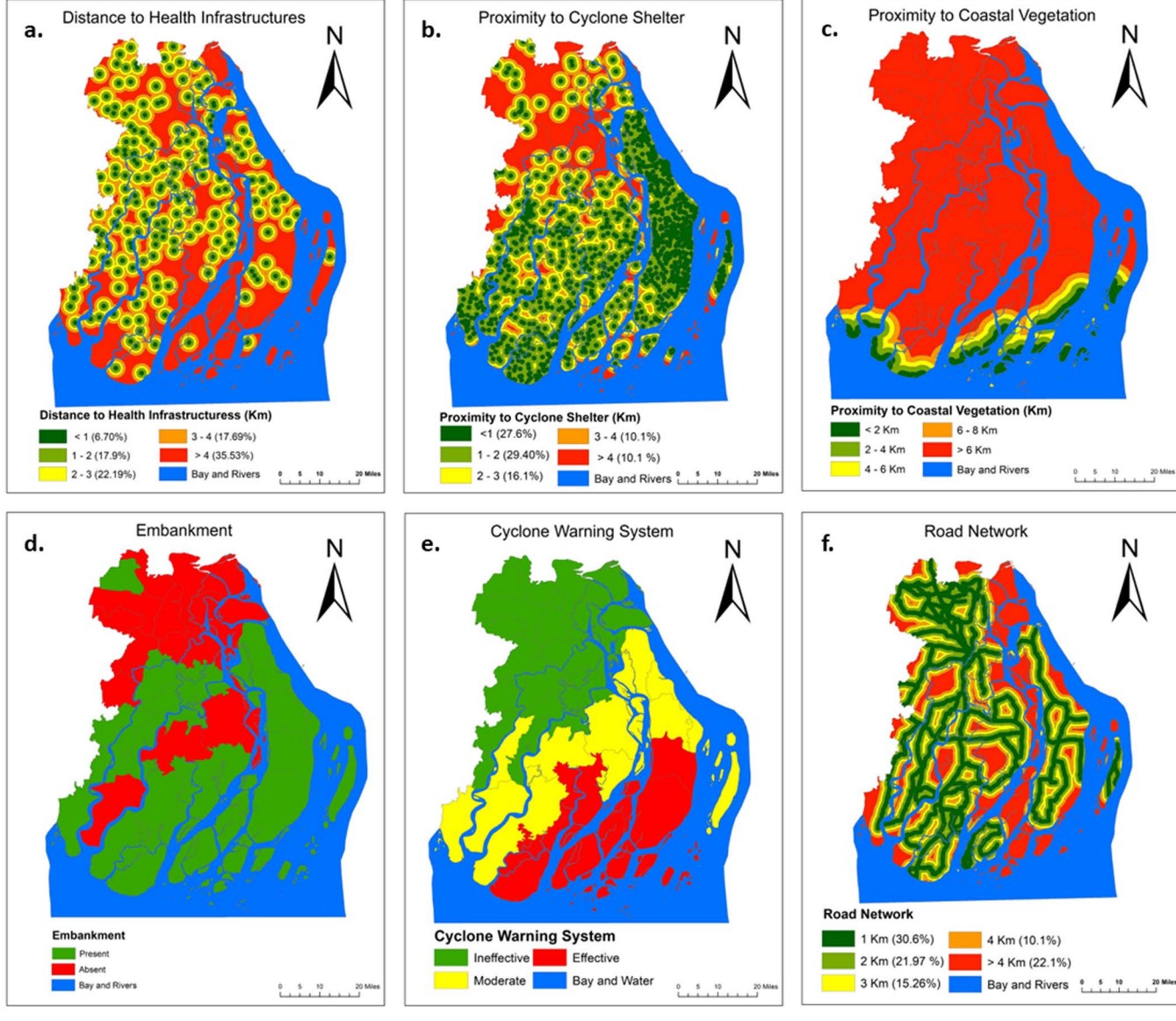

**Fig 5. Features for Mitigation capability.** (a) distance to health infrastructure, (b) distance to cyclone shelter, (c) proximity to coastal vegetation, (d) embankments, and (e) cyclone warning system and (f) road network. [Coastal vegetation map created using Landsat 8 Imagery (30 m resolution) from the U.S. Geological Survey (USGS)].

An early warning system is important to mitigate the damage of a cyclone because it provides details about the position and strength of the cyclone, as well as the expected direction of its travel, increased intensity, coastal districts that are likely to encounter poor weather, as well as recommendations to fishermen, the public, the media, and disaster managers [64]. The warning system data was gathered from the local administrative offices, which included information on the structure, devices, and staff participating in the warning system. Upon conducting field verification of the analyzed data, we proceeded to categorize it into three distinct groups: effective, ineffective, and moderate, in accordance with the established warning system (see Fig 5e).

The presence of a road network plays a crucial role in facilitating relief efforts, rescue operations, and various forms of management measures in the periods preceding, during, and following tropical cyclone occurrences [60]. The data pertaining to the primary roadways was acquired from the Local Government Engineering Department (LGED), and the Euclidean distance tool was employed to generate the spatial representation of the road network (see Fig 5f).

**Ranking of sub-factors.** The study employed two distinct techniques of classification, namely human classification (based on criteria such as closeness to cyclone track and road network) and natural break classification (for elements such as height and slope), in order to assign rankings to the sub-factors. Levels (1–5) signify extremely low vulnerability (rank 1) and very high vulnerability (rank 5), based on the idea that more vulnerability would result in a higher-ranking value (see Table 2). The sub-factors were ranked using block units based on social characteristics. Nevertheless, to provide coherence between the physical and social criteria, the social criteria were ultimately transformed into a raster format while maintaining a comparable cell value. A raster layer with a spatial resolution of 30 m was generated for each of the spatial criteria layers used for ranking. The rank values of spatial criteria layers were then put on a scale of 0–1 by standardizing raster criteria layers. The method of standardization was carried out utilizing Eq. (2).

$$p = \frac{x - min}{-min}$$

(2)

In this context, the variable "x" represents the value assigned to each cell inside the spatial raster layer. The terms "min" and "max" refer to the minimum and maximum values associated with each dataset. Lastly, the variable "p" represents the normalized score.

**Table 2. Feature ranking framework based on the relative contribution to tropical cyclone vulnerability. This table evaluates and ranks individual factors according to their significance in influencing regional vulnerability to tropical cyclones.**

| Factor | Feature | Very Low (1) | Low (2) | Moderate (3) | High (4) | Very High (5) |
|---|---|---|---|---|---|---|
| Physical Vulnerability | Elevation (m) | >18 | 13 - 18 | 9 - 13 | 3 - 9 | < 3 |
| | Slope (%) | 15.58 - 81.08 | 9.54 - 15.58 | 5.72 - 9.54 | 2.86 - 5.72 | 0 - 2.86 |
| | Proximity to Cyclone Track (Km) | NA | 8 - 12 | 4 - 8 | < 4 | NA |
| | Proximity to Coastline (km) | > 80 | 60 - 80 | 40 - 60 | 20 - 40 | < 20 |
| | Land Cover | Bare Soil | Water bodies | Vegetation | Crops | Urban |
| Social Vulnerability | Population Density (per sq. Km) | 247 - 484 | 485 - 763 | 764 - 1103 | 1104 - 1625 | 1626 - 2039 |
| | Children and Older Population (%) | < 48.72 | 48.72 - 52.93 | 52.93 - 54.91 | 54.91 - 56.15 | 56.15 - 58.69 |
| | Disable Population | 972 - 1909 | 1910 - 2732 | 2733 - 4036 | 4037 - 5497 | 5498 - 7531 |
| | Female Population (%) | 47.47 - 49.40 | 49.40 - 50.54 | 50.54 - 51.30 | 51.30 - 51.30 | 51.85 - 52.77 |
| | Literacy Rate (%) | 63.9 - 69.3 | 57.2 - 63.9 | 48.7 - 57.2 | 41.6 - 48.7 | 32.1 - 41.6 |
| | Wooden House (%) | < 12.2 | 12.2 - 57.5 | 57.5 - 86.8 | 86.8 - 93.1 | 93.1 - 97.9 |
| | Agriculture Dependent Population (%) | < 9.18 | 9.18 - 30.31 | 30.31 - 40.96 | 40.96 - 47.47 | 47.47 - 61.27 |
| Mitigation Capacity | Distance to Cyclone Shelter (Km) | <1 | 1 - 2 | 2 - 3 | 3 - 4 | > 4 |
| | Distance to Health Infrastructure (Km) | < 1 | 1 - 2 | 2 - 3 | 3 - 4 | > 4 |
| | Proximity to Coastal Vegetation | < 2 | 2 - 4 | 4 - 6 | 6 - 8 | > 6 |
| | Embankments | NA | Present | NA | Absent | NA |
| | Cyclone Warning System | NA | Effective | Moderate | Ineffective | NA |
| | Road Network (Km) | < 1 | 1 - 2 | 2 - 3 | 3 - 4 | > 4 |

## FAHP and features weighting

In this investigation, an integrated FAHP technique created by Chang was used [65]. Using FAHP, a complex system is broken down into a hierarchy of elements, which typically consists of objectives, evaluation standards, and potential alternatives [66]. In this method, the pair-wise comparison is made simpler using a triangular fuzzy number (TFN), which avoids the need for a laborious comparison process [65].

The initial phase was the selection of relevant criteria for conducting vulnerability assessments of tropical cyclones. In the second phase, pair-wise comparison matrices were constructed using the geometric mean method to integrate the opinions of six experts based on the relative importance of the selected features. The opinions of specialists and those in charge of making decisions are the primary sources that are used to evaluate the criteria in terms of the importance they have in relation to the spatial issue. In the present study, a triangular fuzzy set was employed to facilitate the conversion of language variables into their corresponding quantitative values Eq. (3). The interplay between quantitative values and linguistic elements is depicted in Table 3.

$$R = (a, b, c), \ K = 1, \ 2, \ ....., \ K \tag{3}$$

Where, R = triangular fuzzy member; K = number of DMs; $a = (a_{1 \times} \ a_{2 \times} ........ \times a_{k)}^{k}$, $b = (b_{1 \times} \ b_{2 \times} ........ \times b_{k)}^{k}$, and $c = (c_{1 \times} \ c_{2 \times} ........ \times c_{k)}^{k}$.

In the third, a set of overall priorities for the hierarchy was developed by aggregating and synthesizing the results of pair-wise comparison matrices.

The consistency ratio (CR) Eq. (4) was computed in the fourth phase as a means of providing an explanation for the experts' evaluations in the pair-wise matrices. If the consistency ratio is either equal to or less than 0.1, then the judgment is regarded to be accurate.

$$CR = \text{Consistency Index}/\text{Random Index} \tag{4}$$

The computation of the random index (RI) is performed using the matrix order (n), as defined by Saaty [67]. The statistic known as the consistency index (CI) is calculated using Eq. (5).

$$CI = (\lambda_{max} - n)/(n - 1) \tag{5}$$

Here, $\lambda_{max}$ and n are used to represent the maximum eigenvalue and order of a matrix, respectively [65].

In the fifth stage, the pair-wise matrix criteria weights were converted into linguistic variables with the help of. The determination of the priority weights was carried out using the approach introduced by Chang (see Table 4) [65].

**Table 3. Affiliation function of the scale of linguistics.**

| Variables | Number of Crisp | Triangular Fuzzy Numbers | Reciprocal Triangular Fuzzy Numbers |
|---|---|---|---|
| Strong | 1 | (1,1,1) | (1,1,1) |
| Moderately Strong | 3 | (2,3,4) | (1/4,1/3,1/2) |
| Strong | 5 | (4,5,6) | (1/6,1/5,1/4) |
| Very Strong | 7 | (6,7,8) | (1/8,1/7,1/6) |
| Extremely Strong | 9 | (9,9,9) | (1/9,1/9,1/9) |
| Intermediate | 2 | (1,2,3) | (1/3,1/2,1) |
| | 4 | (3,4,5) | (1/5,1/4,1/3) |
| | 6 | (5,6,7) | (1/7,1/6,1/5) |
| | 8 | (7,8,9) | (1/9,1/8,1/7) |

**Table 4. Weights of features obtained through pairwise comparison matrices, accompanied by the corresponding Consistency Ratio (CR) values to assess consistency.**

| Criteria | Feature | Weight |
|---|---|---|
| Physical vulnerability [Consistency Ratio (CR): 0.03] | Land Cover | 0.130 |
| | Slope (%) | 0.206 |
| | Elevation (m) | 0.210 |
| | Proximity to Cyclone Track (Km) | 0.211 |
| | Proximity to Coastline (km) | 0.209 |
| Social vulnerability [Consistency Ratio (CR): 0.03] | Population Density (per sq. Km) | 0.201 |
| | Female Population (%) | 0.170 |
| | Disable Population | 0.115 |
| | Children and Older Population (%) | 0.1786 |
| | Agriculture Dependent Population (%) | 0.096 |
| | Literacy Rate (%) | 0.0976 |
| | Wooden House (%) | 0.168 |
| Mitigation Capacity | Distance to Health Infrastructure (Km) | 0.054 |
| | Distance to Cyclone Shelter (Km) | 0.233 |
| | Proximity to Coastal Vegetation | 0.201 |
| | Embankments | 0.071 |
| | Cyclone Warning System | 0.169 |
| | Road Network (Km) | 0.196 |
| Overall Vulnerability [Consistency Ratio (CR): 0.01] | Physical vulnerability | 0.660 |
| | Social vulnerability | 0.333 |

## Assessment of vulnerability

When constructing the social, physical, and mitigation capability indices, the overlay analysis was carried out independently, along with the applicable criterion spatial layers and the weights associated with them. Subsequently, the study proceeded to generate the social, physical, and mitigation capabilities maps by categorizing the indices into five distinct groups, spanning from very low to exceedingly high. After that, the study multiplied the physical and social vulnerability indices in order to build a vulnerability index without incorporating the capacity for mitigation. On the contrary, equation Eq. (1) was used to create a vulnerability index that takes into account mitigation capacity. The values of both indices were then normalized so that they could be transformed into a common scale that ranged from 0 to 1, and they were categorized into a number of different susceptibility levels, such as very low, low, moderate, high, and very high.

## Approach to the validation of cyclone vulnerability

The receiver operating characteristics curve (ROC) and the area under the curve (AUC) were utilized in this investigation for the purpose of validating the cyclone vulnerability map that did not include an integrated mitigating capability. The ROC concept has evolved into an established benchmark for assessing the dependability of binary predictors in several domains [68]. The capacity of various statistical methods to discriminate between groups of individuals is being evaluated with the use of ROC curves. These methods integrate a variety of hints, test results, and other data [69]. This method has seen widespread application in evaluating the sensitivity, susceptibility, and risk models of a variety of natural threats, including floods, droughts, and landslides, among others [70]. There is currently no tried-and-true method to validate the tropical cyclone vulnerability map that has been developed [52]. To begin with, the study created an inventory map by marking the regions that had been impacted by prior cyclones. To create the map, the study contains 155 validation sites

based on the findings of previously conducted studies [56], published report ('Emergency Response and Action Plans Interim Report' prepared by Government of People's Republic of Bangladesh 2007 and 'Cyclone Sidr 2008 in Bangladesh' Prepared by the Government of Bangladesh) as well as fieldwork.

## Results

### Spatial distribution of physical vulnerability criteria

Five categories of physical vulnerability to tropical cyclones such as i) LULC ii) slope (%) iii) elevation (m) iv) proximity to cyclone track (km) v) proximity to coastline (km) are used for vulnerability analysis which are shown in Fig 3. Majority of the areas studied are prominently covered by crops and vegetations (see in Fig 3a). These areas are physically vulnerable due to their vulnerability to climate change impacts like flooding, and saline intrusion after cyclones, which can severely disrupt agricultural productivity and ecosystems. Basically, the studies areas are lowland while some portions of Barishal, Jhalakati and some southern portions of the study areas have higher slope (5.72–9.54) (Fig 3b) and elevation (9–18 m) (Fig 3c). The low-lying nature of these areas makes them physically more vulnerable to flooding, and storm surges, exacerbating risks to infrastructure, agriculture and livelihoods. Although parts of southern Borgona, Patuakhali and northern to central Bhola are situated 4–12 km away from cyclone track (Fig 3d), they remain physically vulnerable due to secondary impacts such as storm surges, wind damage, and heavy rainfall induced flooding [20]. Results indicate that Barishal and Jhalakati, being farther from the coastline compared to other areas followed by Fig 3e, are relatively less exposed to direct impacts by cyclones but may still face risks from heavy rainfall, riverine flooding, and secondary effects [2].

### Spatial distribution of social vulnerability criteria

Fig 4 displays the five distinct groups such as i) population density (per sq. km) ii) female population (%) iii) disable population iv) children and older population (%) v) agriculture dependent population (%) vi) literacy rate (%) vii) wooden house (%) that encompass social vulnerability to tropical storms. Results show that the higher population density in northwestern Barishal, Jhalakati, some portions of Borgona and northern Patuakhali (see in the Fig 4a), increases social vulnerability by amplifying exposure to resource scarcity, limited evacuation options and challenges in disaster response and recovery [23]. The lower population density and female population percentage in the southern part may reduce social strain (Fig 4b), but limited social networks could hinder disaster resilience. Conversely, the central (Jhalakati, Patuakhali) and north-western Barishal areas with a higher female population (>50%) (see in the Fig 4b), may face heightened social vulnerability due to gendered disparities in access to resources and decision-making during crises [15]. On the other hand, the higher prevalence of disabled individuals (4037–7531) in Bhola, Patuakhali and Borgona (Fig 4c), coupled with the concentration of children and older population in Barishal, northern Patuakhali and Bhola districts indicates heightened social vulnerability in these areas (Fig 4d). These demographic groups are often more vulnerable to challenges in accessing resources and responding to disasters, necessitating targeted interventions. Additionally, the moderate distribution of vulnerable populations in Borgona, north-eastern Barishal and southern part of Patuakhali underscores the importance of tailored support mechanisms to address diverse vulnerability levels in the region (see in Fig 4d). The heavy reliance on agriculture, with over 48% of the population dependent on it in north-western Borgona and some north-eastern part of Barishal (shown in Fig 4e), increases social vulnerability due to sector's vulnerability to climate change and natural disaster like cyclones. Similarly, the moderate agriculture dominance (41–47%) in Bhola, Pathukhali and Borgona indicates significant dependence on climate-sensitive livelihoods, emphasizing the need for adaptive measures to reduce risk and enhance resilience. The low-literacy rates, below 48%, in the north-eastern (Barishal) and the south-eastern (Bhola, Patukhali) areas amplify social vulnerability as limited literacy restricts access to critical information, resources, and opportunities for livelihood diversification (Fig 4f) [30]. The predominance of wooden houses, over 93% in Borgona and southern Patuakhali, as well as 86.8–93.1% in Bhola, northern Patuakhali, some portions of Borgona and Barishal districts,

indicates high social vulnerability due to the fragility of these structures (see in Fig 4g) [17]. Wooden houses made from wood and tree leaves are particularly susceptible to damage from natural disasters, such as cyclones and floods, increasing the risk of displacement and loss of shelter.

## Spatial distribution of mitigation capacity criteria

In terms of mitigation ability, several factors such as i) distance to health infrastructure (km) ii) distance to Cyclone Shelter (km) iii) proximity to Coastal Vegetation iv) embankments v) Cyclone warning system vi) road network (km) have been used for assessing mitigation capability (Fig 5). The limited proximity to health centers, with over 35% of areas in southern regions like Bhola, Patuakhali and Borgona being far from medical facilities (see in Fig 5a), significantly weakens mitigation capacity in these areas. This lack of access delays emergency medical responses during disasters and hinders routine healthcare, exacerbating vulnerabilities. The presence of cyclone shelters in more than 50% of the southern regions, such as Bhola, Patuakhali and Borgona, reflects a significant mitigation asset (Fig 5b). However, the lack of proper infrastructure limits their effectiveness, reducing their capacity to provide adequate protection during cyclones. The studied areas' distance from the coastline offers some natural protection from direct coastal impacts, but the lack of big trees reduces windbreak capacity and increases exposure to strong winds during cyclones (Fig 5c). The presence of embankments across most of the studied areas enhances mitigation capability by providing critical protection against flooding and storm surges during cyclones (Fig 5d). However, the gaps in embankment coverage in parts of Barishal, Borgona, and Patukhali leave these regions more vulnerable, highlighting the need for extending and maintaining embankment networks [59]. The moderate to effective cyclone warning systems in southern parts of the studied areas significantly enhance mitigation capability by providing early alerts, allowing communities to prepare and evacuate timely (Fig 5e). While the good road network in the studied areas enhances accessibility and evacuation during emergencies, the flooding of these roads during cyclones undermines their effectiveness as a mitigation tool (Fig 5f). To improve mitigation capability, it's essential to develop flood-resistant infrastructure that ensures continuous access during extreme weather events [23].

## Physical, social vulnerability and mitigation capacity

The greater southern parts (especially Bhola, Borgona, Patuakhali) of the studied districts are covered high (more than 10%) and extremely high (nearly 31%) physically vulnerable to the effects of cyclones (shown in Table 5 and Fig 6a). A discernible trend of diminished vulnerability levels, ranging from very low (nearly 5%) to low (more than 23%), is observed in the northern and northwestern regions of Barishal and Jhalakati. This pattern encompasses a very small portion, accounting for about 28% of the total research areas (Table 5 and Fig 6a). To add to it, more than 26% (2370.86 sq. km) of the territory under consideration was classified as moderately vulnerable, and it was positioned in the middle and northern parts of the study areas (Table 5 and Fig 6a).

Fig 6b demonstrates that the southern, southwestern, southeastern, and central areas of the study have a social vulnerability category that ranges from extremely high to high, covering more than 50% of the total studied regions. On the

**Table 5. Spatial distribution of physical and social vulnerability and mitigation capacity in the study area. The aggregated spatial coverage of physical and social vulnerabilities surpasses the area where mitigation capacity is effectively present.**

| Category | Physical vulnerability | | Social vulnerability | | Mitigation capacity | |
|---|---|---|---|---|---|---|
| | Area(km²) | % | Area(km²) | % | Area(km²) | % |
| Very low | 436.17 | 4.86 | 487.37 | 5.74 | 635.19 | 7.23 |
| Low | 2113.49 | 23.55 | 632.45 | 7.45 | 1295.80 | 14.76 |
| Moderate | 2370.86 | 26.42 | 2414.85 | 28.45 | 3549.81 | 40.43 |
| High | 1277.12 | 14.23 | 3191.08 | 37.60 | 1884.55 | 21.46 |
| Very High | 2777.87 | 30.95 | 1761.17 | 20.75 | 1414.74 | 16.11 |

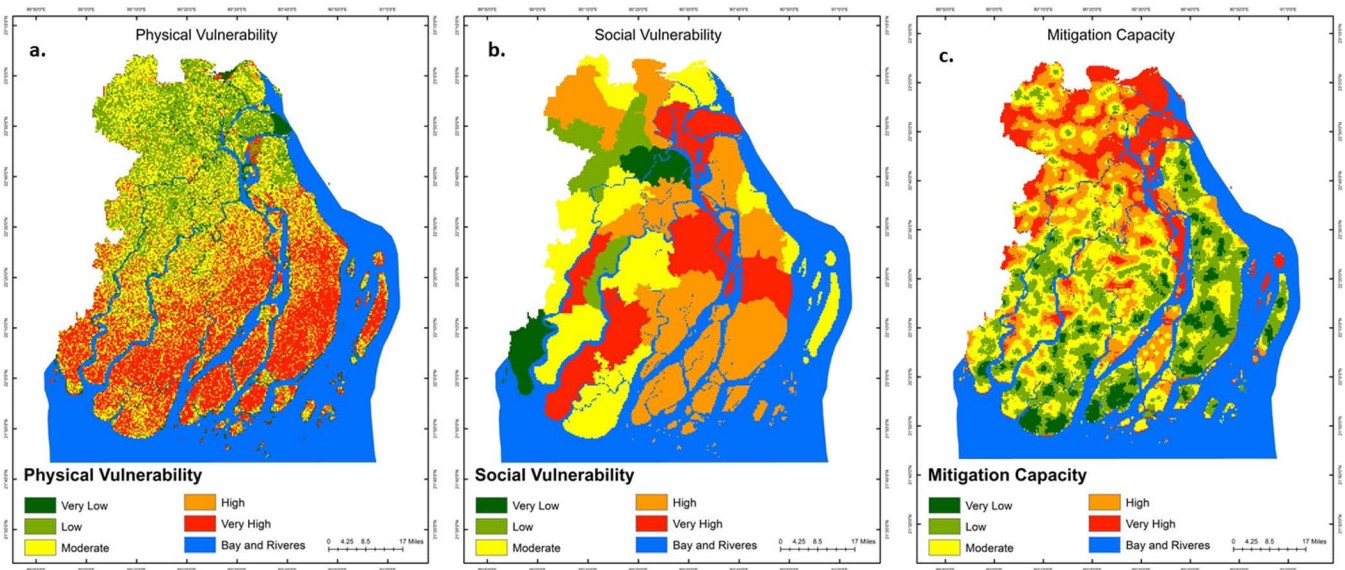

**Fig 6. Figure illustrates the spatial pattern and the level of vulnerability due to tropical cyclones.** (a) physical vulnerability, (b) social vulnerability, and (c) mitigation capacity. [Satellite images courtesy of the U.S. Geological Survey (USGS)].

other hand, some areas in the districts of Barishal, Jhalakati, and Patuakhali, which together encompass around 28% of the study area, are considered to have a moderate level of social vulnerability (Table 5 and Fig 6b). Besides, zones with very low to low social vulnerability cover just above 1119 sq. km of the total area of the region that is researched and can be found in the western and northern parts of the region (see Table 5 and Fig 6b).

The southern and southeastern region have shown very low to low mitigation capacity (shown in Fig 6c) which are covered around 20% areas (Table 5) because of its precarious geographical position and the recurring pattern of being struck by various natural disasters each year. On the other side, the central parts of the districts of Borgona, Patuakhali, and Jhalakati, which account for over 40% of the area surveyed, are considered to have a moderate level of mitigation capacity (see Table 5 and Fig 6c). Most of the Barishal district shows very high to high level of capability (Fig 6c).

### Validation of cyclone vulnerability

Fig 7 illustrates the prediction rate curve, showcasing the model's performance in this research. The AUC for the FAHP model was 0.79, indicating a 79% prediction accuracy. AUC values range from 0.5 to 1.0, with values closer to 1.0 indicating higher accuracy [68,69]. Therefore, the AUC of 79% in this analysis highlights the effectiveness and success of the developed cyclone vulnerability assessment approach, aligning the previous studies [52].

### Discussion

Numerous studies have been conducted to assess cyclone vulnerability at both national and district levels [23,33–35,38,43,52,71]. Most of these studies have focused on evaluating exposure to vulnerability on regional or community scales. However, none have comprehensively considered the key vulnerability dimensions—physical, social, and mitigation capacity—to analyze vulnerability in Bangladesh's coastal regions because of cyclones.

To address this gap, various factors were identified as variables representing physical, social vulnerability and mitigation capacity levels. For example, physical vulnerability factors such as slope, elevation, distance to cyclone tracks, and proximity to the coastline were used as indicators of exposure in a study by [52]. Similarly, sensitivity-related factors, often

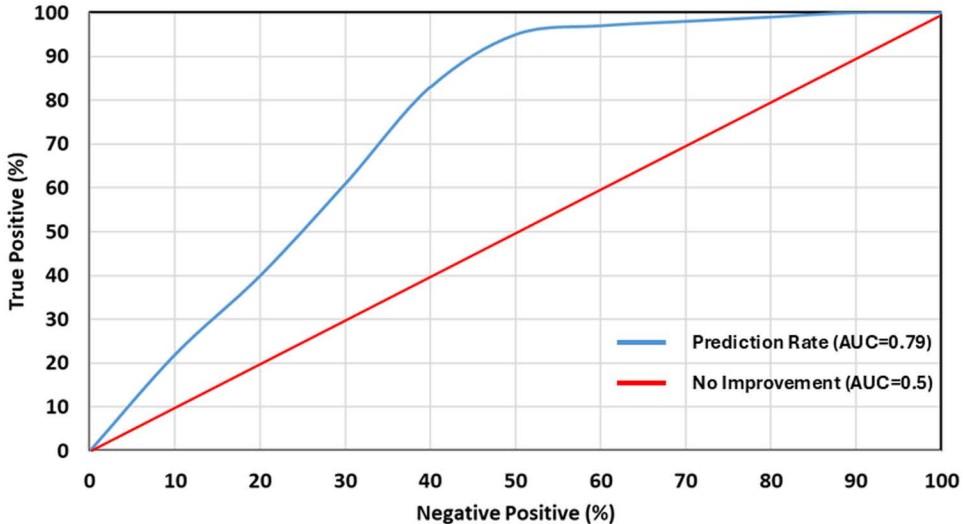

**Fig 7. Area under the curve for prediction rate (79%), indicating a prediction accuracy.** The analysis highlights the effectiveness and success of the developed cyclone vulnerability assessment approaches.

used as social vulnerability indicators in previous research, included elements like major roads and proximity to coastal vegetation, which were also considered as measures of mitigation capacity in the western coastal part of Bangladesh [52]. However, no previous studies are fully considered to assess the vulnerability of the central coastal parts in Bangladesh.

This study evaluated the applicability of selected variables for factor analysis by an integrated FAHP technique using of a triangular fuzzy number (TFN), which avoids the need for a laborious comparison process, confirming their suitability, consistent with findings from other studies [71, 72]. Results show that the consistency ratio (CR) is in between 0.01 to 0.03, aligning with previous studies [52].

The FAHP results revealed that the distance to the coastline and cyclone tract had the strongest and most positive contribution to cyclone vulnerability in terms of physical, supporting with previous studies [34,71]. Highly vulnerable zones included Borgona, Bhola and the southern Patuakhali, while districts like Barishal, and Jhalokati were moderately to less vulnerable. For social vulnerability, seven factors were identified: population Density (per sq. km) (0.201), female population (%) (0.170), disable population (0.115), children and older population (%) (0.179), agriculture dependent population (%) (0.096), literacy rate (%) (0.098) and wooden house (%) (0.168). Among these, the population density had the highest influence. Other factors, such as children & older population and female population, were also significant indicators of social vulnerability, highlighting their role in defining social vulnerability by previous studies [34,71]. In terms of mitigation capacity, regions with close to cyclone shelter and have better cyclone warning systems, such as Barishal, and Jhalokati demonstrated stronger adaptive capacities. Conversely, Borgona, Bhola and Patuakhali, were identified as having very low to adaptive capacity.

The FAHP analytical framework employed in this study, combined with its findings, highlights the importance of distinguishing among physical, social vulnerability, and relationship with mitigation capacity for a more accurate assessment of cyclone vulnerability at the district level [34,52,71]. For instance, while parts of Barishal exhibit higher social vulnerability in terms of demographics (density of population, female population, and agriculture dependent population), their vulnerability remains low due to strong mitigation capacity. Using Equations (1) and (2), the vulnerability of the coastal regions was assessed, revealing that districts such as Borgona, Bhola and Patukhali fall into the very high vulnerability zone. This aligns closely with findings from studies by [34,71]. Additionally, [50,26] identified these districts as highly vulnerable

to coastal flooding caused by cyclones, while [72, 73] reported a high cyclone occurrence ratio in these areas, resulting in more devastating casualties compared to other coastal districts in Bangladesh. Bhola, in particular, has been ranked as the most vulnerable district by [74]. In contrast, [71] categorized Barisal, and Jhalokati as moderate vulnerable in the contexts of whole coastal regions in Bangladesh. This study, however, also identified parts of Barisal, and Jhalokati as being in the moderate to low zone. [74] ranked Barisal as the 7th most at-risk districts, respectively, among Bangladesh's 19 coastal districts.

Understanding the relationship among physical, social vulnerability, and relationship with mitigation capacity is critical for evaluating cyclone vulnerability in Bangladesh's coastal regions. These elements are deeply interconnected, collectively shaping the overall vulnerability of an area. For instance, regions like Borgona, Bhola and the southern Patuakhali, which experience exceptionally high exposure due to their proximity to the coast, face increased physical vulnerability. This geographical exposure is a fundamental factor in assessing potential risks. Rural areas with significant populations of vulnerable groups, such as female, children and the elderly and a high dependency on agriculture, demonstrate increased social vulnerability to cyclone impacts. Factors like disabled population, literacy rate and housing patterns also influence sensitivity, making them crucial indicators of social vulnerability.

Mitigation capacity, the third component, reflects a region's ability to respond to and recover from cyclone events effectively. The interplay between these factors is evident. For example, districts with better warning systems and easy access to cyclone shelters show greater mitigation capacity. Resources like better road networks and health infrastructures bolster community resilience during cyclones, emphasizing their role in mitigating adverse impacts. A comprehensive vulnerability assessment of the coastal zone requires recognizing the complex interactions among these components. For example, a region with low physically vulnerable but high socially vulnerable may still experience elevated vulnerability if its mitigation capacity is insufficient to manage cyclone consequences. Acknowledging these interconnections highlights the intricate and multifaceted nature of vulnerability within Bangladesh's coastal areas.

The detection of coastal vulnerability, physical, social vulnerability, and mitigation capacity using Fuzzy Analytical Hierarchy Process (FAHP) and geospatial techniques represents a relatively novel approach in these central coastal regions. Moreover, comparisons can be drawn between the findings of this study and experiences from neighboring and far countries in the Bay of Bengal region, such as India [43,71,75,76] and China [8,47], which share similar geographical and meteorological characteristics for India and different for China, and face comparable challenges related to cyclones. By drawing these parallels and analyzing regional risks, this research has the potential to serve as a valuable reference for neighboring nations confronting similar issues. The shared experiences and assessments could play a crucial role in fostering regional collaboration and guiding policy-making efforts to mitigate cyclone-related risks. This would enhance vulnerability assessments by providing a framework for taking efficient, targeted actions to reduce cyclone damage, helping policymakers identify the key factors contributing to cyclone destruction.

## Future directions and recommendations

While the present study provides valuable insights into cyclone vulnerability in Bangladesh's central coastal regions, it acknowledges several limitations that may influence the interpretation and generalizability of the findings. The following recommendations and guidelines are proposed to address these constraints and to inform future research in this field:

• **Enhancing primary data collection and field validation**

One of the principal limitations of this study is the heavy reliance on secondary datasets from institutional sources. While these data are generally robust, they may lack real-time accuracy and may not reflect localized socio-environmental

dynamics. Future research should incorporate comprehensive field-based surveys, participatory rural appraisals (PRAs), and community interviews to validate spatial data and to capture community-specific perceptions of vulnerability and adaptive capacity. Ground-truthing of remote sensing output such as land cover classifications, shelter locations, and housing materials—can significantly enhance the reliability of geospatial analyses.

- **Integrating temporal dynamics and longitudinal assessments**

This study offers a static snapshot of vulnerability, which may not adequately reflect the temporal variability of cyclone impacts, particularly under changing climate conditions. Future studies should adopt a longitudinal design to monitor changes in vulnerability patterns over time. This can be achieved by using time-series remote sensing data, recurring census information, and historical cyclone damage assessments to better understand trends, resilience-building, and degradation over the years.

- **Expanding the multi-scale analytical framework**

The current district-level analysis, while useful for regional planning, may obscure important micro-level disparities in vulnerability. Future work should explore multi-scale vulnerability assessments, combining national, district, union, and community-level resolutions. This would allow for the detection of localized vulnerability hotspots, particularly within urban slums, embankment-fringe communities, or isolated chars (river islands), which are often marginalized in aggregated data.

- **Incorporating complex social variables and behavioral indicators**

Social vulnerability is inherently complex and deeply context-specific. Future studies should aim to integrate qualitative social indicators, such as household coping mechanisms, trust in early warning systems, disaster preparedness culture, gender dynamics, and governance quality. These indicators provide a more nuanced understanding of vulnerability beyond structural indicators like population density or literacy rates. Behavioral data, when combined with spatial information, can reveal important insights into the effectiveness of mitigation interventions.

- **Strengthening the modeling approach**

While FAHP offers a structured and effective method for prioritizing vulnerability criteria, it assumes linear relationships and relies on expert judgment, which introduces potential subjectivity. Future research could explore the use of hybrid models, such as FAHP integrated with machine learning techniques (e.g., Random Forest, SVM, or ANN) to enhance classification accuracy and objectivity. The application of Bayesian models or agent-based simulations could also capture complex interdependencies and human responses to cyclones in dynamic settings.

- **Addressing climate change and sea-level rise projections**

Although this study incorporates historic cyclone data, it does not fully consider future cyclone intensity or sea-level rise scenarios projected by climate models. Future vulnerability assessments should include climate change projections, scenario-based analyses (e.g., RCP pathways), and probabilistic hazard mapping to better account for long-term risks and evolving exposure profiles in the coastal region.

- **Enhancing regional and international collaboration**

Given the shared vulnerability of Bay of Bengal nations, future research should promote cross-border comparative studies involving India, Sri Lanka, and Myanmar. Developing a regional framework for vulnerability assessment—supported by standardized indicators and data sharing protocols—could facilitate knowledge exchange, mutual learning, and the design of regionally coherent disaster management policies.

## Conclusion

This study presented a comprehensive assessment of cyclone vulnerability in the central coastal region of Bangladesh, encompassing an area of 8,975.51 km². Through the integration of the Fuzzy Analytical Hierarchy Process (FAHP) and geospatial techniques, we developed spatially explicit vulnerability maps that accounted for physical, social, and mitigative dimensions. The validation of the model using ROC and AUC metrics confirmed the reliability of our approach. Findings revealed that southern Bhola, Barguna, and Patuakhali—including dispersed island communities—are at very high risk from tropical cyclones due to a combination of low elevation, high population density, poor infrastructure, and limited access to emergency services. Interestingly, even inland areas such as parts of northeastern Barishal were found to be highly vulnerable, owing to their insular geography and exposure to major river systems.

Despite limitations in data quality—such as outdated census figures, medium-resolution remote sensing inputs, and inconsistently updated infrastructure datasets, the methodological framework proved effective for synthesizing diverse risk factors. Field verification helped address some of the data gaps, but future work must prioritize access to high-resolution, real-time datasets and consistent updates from local agencies.

Crucially, this study offers more than a diagnostic tool—it provides actionable insights to inform disaster risk reduction (DRR) strategies and climate-resilient policy planning. The study findings highlight the need for targeted investments in cyclone shelters, embankments, early warning dissemination systems, and healthcare infrastructure, especially in highly exposed zones. Furthermore, promoting nature-based solutions, such as coastal afforestation and mangrove restoration, can serve as buffers against storm surges while enhancing ecological resilience.

By aligning vulnerability assessments with regional development planning, the outcomes of this study can directly support the Bangladesh Delta Plan 2100, the National Plan for Disaster Management (NPDM), and the Sendai Framework for Disaster Risk Reduction (2015–2030). Additionally, the study contributes to Sustainable Development Goals (SDGs), especially Goal 11 (Sustainable Cities and Communities) and Goal 13 (Climate Action), by offering insights that can help build resilience among vulnerable coastal populations. This research also offers a transferable methodology for other cyclone-prone coastal regions globally, where socio-environmental fragility and climate risks intersect.

Ultimately, this study underscores the urgency of integrating spatial vulnerability assessments into proactive policy frameworks that prioritize both structural adaptation and social equity. Doing so will not only enhance community preparedness and reduce cyclone-induced losses but also support the long-term sustainability of coastal livelihoods in Bangladesh and beyond.

## Acknowledgments

The authors are grateful to the University of Barishal, the Survey of Bangladesh (SoB), Bangladesh Bureau of Statistics (BBS), Local Government Engineering Department (LGED), the United States Geological Survey (USGS), and International Best Track Archive for Climate Stewardship (IBTrACS), for providing the essential datasets.

## Author contributions

**Conceptualization:** Md Abdullah Salman.

**Data curation:** Mahmudul Hasan Rakib, Shacin Chandra Saha, Md Emdadul Haque.

**Investigation:** Mahmudul Hasan Rakib, Shacin Chandra Saha, Md Emdadul Haque.

**Methodology:** Md Abdullah Salman.

**Project administration:** Md Abdullah Salman.

**Software:** Mahmudul Hasan Rakib, Shacin Chandra Saha, Md Emdadul Haque, Md Sabbir Hossen.

**Supervision:** Md Abdullah Salman.

**Writing – original draft:** Md Abdullah Salman, Mahmudul Hasan Rakib.

**Writing – review & editing:** Md Abdullah Salman, Md Sabbir Hossen.

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
