## [Editor Report · Decision Letter 0]

Dear Dr. Salman,

Thank you for submitting your manuscript to PLOS ONE. After careful consideration, we feel that it has merit but does not fully meet PLOS ONE’s publication criteria as it currently stands. Therefore, we invite you to submit a revised version of the manuscript that addresses the points raised during the review process.

We look forward to receiving your revised manuscript.

Kind regards,

Rukhsana Rukhsana

Academic Editor

PLOS ONE

5. We note that Figures 1, 3A-E, 4A-G, 5A-F, 6, 7, 8 in your submission contain [map/satellite] images which may be copyrighted. All PLOS content is published under the Creative Commons Attribution License (CC BY 4.0), which means that the manuscript, images, and Supporting Information files will be freely available online, and any third party is permitted to access, download, copy, distribute, and use these materials in any way, even commercially, with proper attribution. For these reasons, we cannot publish previously copyrighted maps or satellite images created using proprietary data, such as Google software (Google Maps, Street View, and Earth). For more information, see our copyright guidelines: http://journals.plos.org/plosone/s/licenses-and-copyright.

1. You may seek permission from the original copyright holder of Figures 1, 3A-E, 4A-G, 5A-F, 6, 7, 8 to publish the content specifically under the CC BY 4.0 license. 

Natural Earth (public domain): http://www.naturalea

Additional Editor Comments:

Review Report.

1. The abstract could benefit from a more streamlined presentation. Some sentences are slightly repetitive, particularly regarding vulnerability factors, and could be combined for brevity. he abstract could more explicitly highlight the practical implications of the study's findings, particularly in terms of how the results could influence policy or lead to tangible improvements in cyclone preparedness.

2. "Spatial Distribution of Vulnerability" Section in results and discussion, The section is generally clear but could be structured better for readability. The discussion on physical, social, and mitigation vulnerability seems fragmented. It might help to more clearly separate the findings on each criterion (physical, social, and mitigation), rather than switching between them in the same paragraph.

3. When referring to maps and tables, ensure a consistent description (e.g., "Fig 6 illustrates..." followed by "As shown in Table 5...") to improve flow.

4. The mention of figures (Fig 6, Fig 7, Fig 8) and tables is helpful but would benefit from a clearer connection between the text and the visuals. It might be more effective to explicitly state what each figure or table illustrates before or after mentioning it.

5. Ensure that the references to the figures and tables are numbered and consistently formatted.

6. The geographical locations mentioned (e.g., Bhola, Barguna, Patuakhali) are important but need further clarification regarding why these areas are specifically vulnerable (e.g., do they have a higher proportion of vulnerable populations, lack of infrastructure, or other unique factors?).

7. The percentages of the total area mentioned in relation to vulnerability categories are useful, but further explanation of how these figures were derived would enhance credibility. For instance, were these areas assessed based on a specific methodology (e.g., satellite data, field surveys)?

8. Resulte and discussion shoulf be separately explained in more detail.

9. The terms used to describe vulnerability (e.g., “very high,” “high,” “moderate,” etc.) are appropriate, but consistency in terminology is key. Ensure that terms like "susceptibility," "vulnerability," and "sensitivity" are used clearly and consistently across the entire report to avoid confusion.

10. The use of "moderate" and "moderately vulnerable" seems a bit repetitive, as these terms are used in both physical and social vulnerability contexts. Clarifying how they differ across contexts would improve precision.

11. The data on the physical vulnerability categories (e.g., “high” and “very high” vulnerability accounting for 45% of the area) is helpful, but the underlying factors (e.g., land cover, elevation, proximity to the coast) should be explained in more detail to give readers a clearer understanding of why these factors contribute to vulnerability.

12. Similarly, for social vulnerability, while the description of factors such as population density, gender, disability, and agricultural reliance is good, a more detailed discussion on how these factors contribute to vulnerability would strengthen the analysis.

13. Mitigation Capacity Discussion:The section on mitigation capacity could benefit from a more nuanced analysis. While it mentions "poor road networks," "poor cyclone warning systems," and "lack of infrastructure," it would be useful to describe the extent of these deficiencies. For example, how do these issues compare across regions? Are some areas better equipped despite the overall low mitigation capacity in the southern and southeastern regions?

14. Additionally, while the findings about Barishal district having higher mitigation capacity are noted, the factors contributing to this higher capacity (e.g., better infrastructure or preparedness) could be explored more in-depth.

15. • The percentages and area calculations (e.g., "30.95% very high vulnerability", "26.42% moderate vulnerability") are useful, but they need clearer contextual explanation. A brief mention of how these areas were classified based on the vulnerability factors would add value. Were the classifications based on specific thresholds or ranges?

16. For mitigation capacity, the percentages are helpful, but more detailed descriptions of what "very low," "moderate," or "high" mitigation capacity entails in practice would provide better insights into the actual conditions on the ground.

17. High resolution figure should be given.
---

## [Author Response · Author response to Decision Letter 1]

7 Jan 2025

Dear Editor & Reviewer,

Thanks for your valuable feedback for improving my manuscript. According to Journal's requirement and editor's comments, I have revised all the required sections.

Looking forward to hearing the acceptance of my manuscript.

Thanks,

Md Abdullah Salman

---

## [Decision Letter · Decision Letter 1]

Dear Dr. Salman,

We look forward to receiving your revised manuscript.

Kind regards,

Sher Muhammad, PhD

Academic Editor

PLOS ONE

**Additional Editor Comments:**

Thank you for submitting your manuscript to PLOS ONE. We’ve received reports from two anonymous reviewers, both of whom recommend major revisions. In light of their detailed feedback, please prepare and submit a substantially revised version addressing reviewers' comments.

Reviewers' comments:

Reviewer's Responses to Questions

**Comments to the Author**

Reviewer #1: (No Response)

Reviewer #2: All comments have been addressed

2. Is the manuscript technically sound, and do the data support the conclusions?

Reviewer #1: (No Response)

Reviewer #2: Yes

3. Has the statistical analysis been performed appropriately and rigorously?

Reviewer #1: (No Response)

Reviewer #2: Yes

4. Have the authors made all data underlying the findings in their manuscript fully available?

Reviewer #1: (No Response)

Reviewer #2: No

5. Is the manuscript presented in an intelligible fashion and written in standard English?

Reviewer #1: (No Response)

Reviewer #2: Yes

**Reviewer #1: ** This study employs FAHP and geospatial techniques to assess tropical cyclone vulnerability in central coastal Bangladesh. The manuscript is well written and structured and the presented findings provide actionable insights for policymakers and NGOs to improve cyclone preparedness and resilience in vulnerable coastal communities. Few adjustments are needed before being considered for publication, and I would be ready to assess the revised version again after applying the following adjustments:

Limit the use of first person pronouns (we used, we utilized, our study…etc.) across the whole manuscript.

Line 123-125: You stated that “According to the literature, the Analytical Hierarchy Process (AHP) is the most prevalent and well-suited approach for spatial multi criteria evaluation [36].” Add that AHP is widly recognized in risk assessment associated with cyclones. Add a relevant reference such as “Mohseni, U., Jat, P. K. and Siriteja, V. (2025). Multi-criteria analysis-based mapping of the cyclone-induced pluvial flooding in coastal areas of India. DYSONA - Applied Science, 6(2), 309-321. https://doi.org/10.30493/das.2025.490282”

Line 155-157: You stated that “Super Cyclones Sidr (2007) and Aila (2009) have severely damaged the coast, killing 3,500 people, injuring 191, and causing significant environmental and property damage [30].” Add that many still perceive them as a major cause of socioeconomic vulnerability and water scarcity in Southwestern Bangladesh. Add a relevant reference such as “Bhowmik, D., Kader, Z., Hosen, B. and Hossain, S. (2025). Exploring socioeconomic vulnerability and natural disasters impacts on water access in the south coastal region of Bangladesh. DYSONA - Applied Science, 6(1), 186-199. https://doi.org/10.30493/das.2024.483646”

Fig. 2 is old fashion and somewhat mundane. Try making it more compact and modern adding some light coloring to it

The results are well-presented and discussed. You also included a limitation section. However, you should discuss your view on how to overcome these limitation, and set few guidelines for future studies in light of your debates.

Enhance the captions of figures and tables. A good caption should include all the data needed for a reader to comprehend the general purpose of figures and tables without returning to the text.

**Reviewer #2:**  Title of the research: Cyclone vulnerability assessment of the central coast of Bangladesh: a comprehensive study utilizing FAHP and geospatial techniques

The coastal belt of Bangladesh is among the most cyclone-prone regions in South Asia. This research proposes an advanced mapping framework to assess tropical cyclone vulnerability in the central coastal zone using a combination of geospatial analysis and the Fuzzy Analytical Hierarchy Process (FAHP). The study evaluates the spatial distribution of cyclone risk by incorporating eighteen indicators that span physical exposure, social sensitivity, and mitigation capacity. The analysis reveals that the southern fringe districts—particularly Bhola, Barguna, and Patuakhali—are at heightened risk. These areas are characterized by frequent cyclone occurrences, close proximity to the Bay of Bengal, low-lying terrain, gentle slopes, and populations with high dependency ratios, including women, persons with disabilities, and agricultural workers. The study also identifies significant weaknesses in existing mitigation infrastructure, including limited access to cyclone shelters, inadequate warning systems, and sparse road connectivity.

The findings offer valuable insights for policymakers, development agencies, and local authorities aiming to design targeted interventions and strengthen disaster risk reduction mechanisms in vulnerable coastal zones. While I support the potential of this study for publication, I recommend a major revision based on the outcomes of the peer review process to enhance clarity, rigor, and policy relevance.

Justification of research: numerous studies have been conducted on cyclone vulnerability assessment in the region. It is better to state what is the new contribution from this study at the outset of this study.

Literature review and results: The study consults limited international literature of the discipline. Relevant examples can be drawn from the following recent sources in disaster prone settings:

• Change in cyclone disaster vulnerability and response in coastal Bangladesh

• Factors of cyclone disaster deaths in coastal Bangladesh,

• Reasons for non-evacuation and shelter-seeking behaviour of local population following cyclone warnings along the Bangladesh coast

Results and Discussion:

Since the results and discussion are presented together without a clear distinction, the interpretation of the data and the broader implications of the study remain ambiguous. This structure makes it difficult for readers to differentiate between the empirical findings and the authors' analytical insights or contextual explanations. As a result, the contribution of the study to existing knowledge, as well as its practical significance for cyclone vulnerability assessment and mitigation planning, is not effectively communicated. To enhance clarity and impact, the authors should consider separating the results from the discussion or, at a minimum, ensuring that each key finding is followed by a focused interpretation and explanation of its relevance within the broader research context. This would allow readers to better understand how the data supports the study’s conclusions and recommendations.

It is also less clear how the study will align with national and international development agendas. Conclusions seem to be just repeating some findings without linking the research findings with relevant disaster management policy and strategies in the region.

**Do you want your identity to be public for this peer review?** For information about this choice, including consent withdrawal, please see our Privacy Policy

Reviewer #1: No

Reviewer #2: **Yes: ** Edris Alam

---

## [Author Response · Author response to Decision Letter 2]

29 May 2025

Hello Editor & Reviewers,

Your required comments have revised and re-submitted.

Thanks,

Abdullah

---

## [Decision Letter · Decision Letter 2]

Cyclone vulnerability assessment of the central coast of Bangladesh: a comprehensive study utilizing FAHP and geospatial techniques

PONE-D-24-47688R2

Dear Dr. Salman,

We’re pleased to inform you that your manuscript has been judged scientifically suitable for publication and will be formally accepted for publication once it meets all outstanding technical requirements.

Kind regards,

Sher Muhammad, PhD

Academic Editor

PLOS ONE

Additional Editor Comments (optional):

Thank you for considering PLOS One for your manuscript. I am glad to recommend your manuscript for publication.

Reviewers' comments:

Reviewer's Responses to Questions

**Comments to the Author**

Reviewer #1: All comments have been addressed

Reviewer #2: All comments have been addressed

2. Is the manuscript technically sound, and do the data support the conclusions?

Reviewer #1: Yes

Reviewer #2: Yes

3. Has the statistical analysis been performed appropriately and rigorously?

Reviewer #1: Yes

Reviewer #2: Yes

4. Have the authors made all data underlying the findings in their manuscript fully available?

Reviewer #1: Yes

Reviewer #2: Yes

5. Is the manuscript presented in an intelligible fashion and written in standard English?

Reviewer #1: Yes

Reviewer #2: Yes

Reviewer #1: ................................................................................................................

Reviewer #2: The authors have demonstrated commendable responsiveness to the previous review comments. The revised manuscript clearly reflects the incorporation of suggested improvements, especially in terms of methodological clarity and explanation of the FAHP process. I appreciate the authors’ efforts in addressing all previous concerns thoughtfully and thoroughly. The revised manuscript represents a significant improvement and stands as a strong, publishable contribution to the field of disaster risk assessment and geospatial analysis.

**Do you want your identity to be public for this peer review?** For information about this choice, including consent withdrawal, please see our Privacy Policy

Reviewer #1: No

Reviewer #2: **Yes: ** Edris Alam

---

## [Editor Report · Acceptance letter]

PONE-D-24-47688R2

PLOS ONE

Dear Dr. Salman,

I'm pleased to inform you that your manuscript has been deemed suitable for publication in PLOS ONE. Congratulations! Your manuscript is now being handed over to our production team.

Kind regards,

on behalf of

Dr. Sher Muhammad

Academic Editor

PLOS ONE